# The Association of Glucocorticosteroid Treatment with WBC Count in Patients with COPD Exacerbation

**DOI:** 10.3390/jcm8101697

**Published:** 2019-10-16

**Authors:** Amit Frenkel, Eric Kachko, Victor Novack, Moti Klein, Evgeni Brotfain, Leonid Koyfman, Nimrod Maimon

**Affiliations:** 1Intensive Care Unit, Soroka University Medical Center, Beer Sheva 8410101, Israel; motik@clalit.org.il (M.K.); bem1975@gmail.com (E.B.); LeonidKo@clalit.org.il (L.K.); 2Faculty of Health Sciences, Ben-Gurion University of the Negev, Beer Sheva 8410101, Israel; a.kachko@gmail.com (E.K.); VictorNo@clalit.org.il (V.N.); nimrod.maimon@clalit.org.il (N.M.); 3Clinical Research Center, Soroka University Medical Center, Beer Sheva 8410101, Israel; 4Division of Medicine, Soroka University Medical Center, Beer Sheva 8410101, Israel

**Keywords:** glucocorticosteroids (GCS), chronic obstructive pulmonary disease (COPD), leukocytosis, eosinophils

## Abstract

Systematic glucocorticosteroids (GCS) are used to treat chronic obstructive pulmonary disease (COPD) and can cause leukocytosis. Distinguishing the effect of GCS on leukocyte level from infection-induced leukocytosis is important. We sought to quantify the effect of chronic GCS treatment on leukocytosis level in patients with COPD exacerbation. We reviewed the records of patients with COPD exacerbation and fever hospitalized in a tertiary medical center in 2003–2014. Patients were classified according to the GCS treatment they received: chronic GCS treatment (CST), acute GCS treatment (AST), and no prior GCS treatment (NGCS). We used the eosinophil absolute count as a marker of compliance and efficacy of steroid treatment. The primary outcome was the maximal white blood cell (WBC) count within the first 24 h of admission. Of 834 patients, 161 were categorized as CST, 116 AST, and 557 NGCS. The overall maximal leukocyte count was higher and the eosinophil count lower in the two GCS therapy groups. In patients with COPD exacerbation and fever, acutely treated with GCS, the mean increase in the WBC count was more evident when the eosinophils were undetectable (absolute count of zero). This supports leukocytosis level as a marker of disease course in COPD and fever.

## 1. Introduction

Treatment with systematic glucocorticoids (GCS) is occasionally a component of the therapeutic regimen of chronic obstructive pulmonary disease (COPD) [1]. Due to the wide range of side effects of GCS, the therapeutic strategy is usually conservative, and treatment is reserved for periods of exacerbation. Nevertheless, for some patients, chronic use of steroids is the only solution that affords disease stabilization [2].

GCS are known to increase the number of white blood cells (WBC), a phenomenon called “glucocorticoid induced leukocytosis”. The increase is mainly in the neutrophilic cells, while the count of the other circulating cells, especially the eosinophils, decreases [3]. Indeed, previous studies used blood eosinophil count as a marker for GCS treatment, for patient compliance and for titration of GCS therapy [4].

The level of leukocytes is a key parameter in the clinical assessment of patients with COPD exacerbation, with and without fever. The presentation of high WBC count makes it important to distinguish the effect of the GCS from the effect of the possible infection. Other studies of steroid-induced leukocytosis in patients with acute infections [5,6] did not focus specifically on patients with fever and acute exacerbation of COPD. Therefore, the main purpose of this study was to quantify the effect of steroid treatment on the level of leukocytosis in patients hospitalized with fever and acute exacerbation of COPD.

## 2. Methods

### 2.1. Study Population

In this retrospective cohort study, we assessed all COPD patients admitted to Soroka University Medical Center with exacerbation of their disease during the years 2003–2014. We included patients who were over 18 years old, with a core body temperature of 38 °C or above, with COPD exacerbation as their main diagnosis. We excluded from the analysis all patients with oncologic or hematologic diseases.

### 2.2. Steroid Treatment

Three groups of patients were compared in this study, according to treatment with GCS: acute, chronic, and no GCS therapy (NGCS). Chronic GCS treatment (CST) was defined as a daily oral dose or an alternate-day oral dose of 5 mg or more of prednisone, for a period of at least 28 days. Acute GCS treatment (AST) was defined as a daily oral dose or an alternate-day oral dose of 5 mg or more of prednisone, for less than 28 days.

The primary outcome was the maximal WBC count, and the secondary outcome was the maximal neutrophil (PMN) count. Both outcomes were measured within the first 24 h from hospital admission and were compared to the minimal eosinophil count and maximal fever (°C).

### 2.3. Statistical Analysis

Normally distributed variables were described by means ± standard deviations; and other distributions by medians and interquartile range (IQR). Categorical variables were summarized by frequencies of available cases. For between covariate associations, the *t*-test or ANOVA test were used for the comparison of normally distributed continuous variables and the Mann–Whitney or Kruskal–Wallis tests for those not normally distributed. Chi-Square or Fisher’s exact test were used for the comparison of dichotomous variables. Data were summarized using frequency tables, summary statistics, confidence intervals, and *p*-values. Associations of WBC and PMN counts with the eosinophil count and body temperature were analyzed by fitting linear curves.

Multivariate analyses for WBC and PMN counts were conducted using a linear regression model. The models were first fitted in the NGCS group. The variables were introduced to the model based on clinical and statistical significance (*p*-value < 0.05 in univariate analysis). The final models included the following variables: age, gender, and degree of fever.

Using the linear regression equation, the expected WBC count and expected PMN count were calculated for each patient in both the AST and CST groups. For the steroid groups, differences between the observed and expected values of WBC and PMN counts (calculated based on the NGCS group regression model) were shown using linear curves as a function of the minimal eosinophil count and body temperature.

All statistical analyses were conducted with a two-sided significance level of α = 0.05 and 95% confidence interval. The statistical analysis was performed by the Statistical Package for the Social Sciences (IMB SPSS STATISTICS, version 20, IBM, Armonk, NY, USA)

Ethics: The work was conducted with the formal approval of the local human subjects committees at Soroka University Medical Center, Israel, Ethics approval number 0304-15(SUMC).

## 3. Results

### 3.1. Patient Population

The study comprised 834 patients diagnosed with COPD exacerbation and fever: 161 in the CST group, 116 in the AST group, and 557 who were not treated with GCS (NGCS). Table 1 summarizes characteristics of the study population: The oldest mean age was noted in the CST group (71.7 ± 11.0 years). The majority of patients in all three groups were current or former smokers: 77.0% in the CST, 74.1% in the AST, and 69.8% in the NGCS. The majority of the patients were males. The proportion of patients with type 2 diabetes was significantly higher in the CST group (57.8%) than in the other two groups (40.5% in the AST group, 39.0% in the NGCS group, *p* < 0.001). The most prevalent diagnosis, in addition to COPD, was pneumonia (33.5% of CST, 37.1% of AST, and 37.5% of the NGCS group). The proportion of patients with bacteremia was higher in the NGCS group (4.7%) than in the AST (2.6%) and CST groups (1.9%). The maximal body temperature measured during hospitalization was similar in all three groups: the NGCS group (38.6 ± 0.6 °C), the AST group (38.5 ± 0.4 °C), and the CST group (38.5 ± 0.5 °C).

### 3.2. White Blood Cell Count 

Baseline laboratory tests of the patients prior to the current illness (Table 2) showed, as expected, that the mean WBC count was significantly higher in the two GCS treatment groups (10.0 ± 5.5 × 10^9^ /L in the AST and 10.2 ± 2.9 × 10^9^/L in the CST) compared to the NGCS group (8.5 × 10^9^/L ± 2.6). During hospitalization, the mean maximal leukocyte count was also higher in the GCS treatment groups than in the NGCS group: 16.1 ± 9.8 × 10^9^/L in the AST group, 14.2 ± 6.9 × 10^9^ /L in the CST group, and 12.1 ± 6.2 × 10^9^ /L in the NGCS group (*p* < 0.001). The eosinophil count was suppressed more in the GCS groups than in the NGCS group: mean eosinophil count of 0.08 ± 0.09 × 10^9^ /L in the AST group, 0.05 ± 0.07.5 × 10^9^ /L in the chronic GCS treatment and 0.11 ± 0.14 × 10^9^/L in the NGST group (*p* < 0.001).

The mean neutrophil count was also higher in both GCS treatment groups than in the NGCS group: 13.1 ± 6.8 × 10^9^/L in the AST group, 12.2 ± 5.4 × 10^9^/L in the CST group, and 10.8 ± 5.8 × 10^9^/L in the NGCS group (*p* < 0.001). WBC count was higher for the two treatment groups than for the NGCS group at all body temperatures; values increased with temperature, particularly for the CST group (Figure 1). The trends for maximum neutrophil values as a function of body temperature were similar to those of maximum WBC count as a function of body temperature (data not shown). The difference increases with temperature and reaches 6 × 10^9^/L at 40.2 °C. A multivariable linear regression analysis showed significantly elevated WBC counts after adjustment for age, smoking, maximal fever, pneumonia, and sepsis, which were similar in the two GCS treatment groups (Table 3). The R square value was 0.76, and the adjusted R square value 0.68.

### 3.3. Eosinophil Count as a Marker of Glucocorticoid Effect

Among patients treated with steroids and with undetectable absolute eosinophil count (undetectable eosinophils), the mean difference between the expected and the observed WBC count was 5.1 ± 12.7.5 × 10^9^/L for the AST and 2.8 ± 7.4 × 10^9^/L for the CST groups. By contrast, for patients who were treated with GCS and had a detectable eosinophil count (absolute count higher than zero), the mean differences between the expected and observed WBC counts were 1.5 ± 5.7 × 10^9^/L for the AST and 0.8 ± 5.7 × 10^9^ /L for the CST (Table 4 and Table 5).

## 4. Discussion

The main finding of this study is that in patients with COPD exacerbation, both chronic and acute GCS therapy was associated with an independent increase in the WBC count to the level of 3–5 × 10^9^/L. This finding was more evident in patients with undetectable levels of eosinophils.

According to a systemic review published in 2015 [7], the majority of the clinical guidelines recommend leukocytosis as a standalone criterion for administering antibiotic treatment in patients with COPD exacerbation. Since the level of leukocytosis is influenced by the infection process [8] and the physiologic stress response [9], as well as the steroid effect, distinguishing the degree of effects of these factors is crucial. Therefore, based on our results, in patients with acute or chronic treatment with GCS, we recommend subtracting 3–5 × 10^9^/L from the actual WBC count to obtain an unbiased, nonsteroid-induced estimate. We emphasize that physicians should consider all clinically available data in deciding whether to administer antibiotic treatment and not rely solely on the level of leukocytosis.

Similarly, the ultimate duration of antibiotic therapy in COPD patients is unclear [10] and usually ranges from 5 to 10 days in most clinical guidelines. Since it is reasonable to assume that the degree of leukocytosis serves as only one of the parameters that guides clinicians in the decision to cease antibiotics therapy, the same approach of subtracting 3–5 × 10^9^/L from the actual WBC count seems appropriate also here.

Blood eosinophil has been proposed as a personalized biomarker-guide to reduce systemic corticosteroid exposure [11]. The issue investigated in the current study is different. Since peripheral blood eosinophils are known to be suppressed by systemic GCS [3], we considered if blood eosinophil level can serve as a direct marker of the effect of GCS. Indeed, we found that the increase of WBC in patients with COPD exacerbation treated with chronic or acute GCS was more evident in patients with undetectable levels of eosinophils. Thus, we assume that mild elevation of WBC in patients with COPD exacerbation treated with GCS, together with normal levels of eosinophils, can be a clue for patient noncompliance or a clue for an inadequate effect of GCS due to other reasons, such as malabsorption, patient error in medical consumption, and vomiting. Moreover, such an eosinophil level can represent a coexisting disease, for example, atopic dermatitis, inflammatory bowel disease, lymphoma, leukemia or drug allergy.

In this cohort study, more than 60% of our COPD patients were males, and the mean age was 69 years. These characteristics are similar to those of COPD patients in a general population in the Netherlands: about 57% in that study were males, and the mean age was 67.8 + −12.1 years [12].

The novelty of this study is its focus on patients with fever and acute exacerbations of COPD. The current study has a number of limitations. First, it included only 834 patients with COPD exacerbation and fever, which is considered a relatively small retrospective study, from a single center. Second, GCS are known to have an antipyretic effect [13]. This would have excluded from the analysis normothermic patients with COPD exacerbation who were treated with GCS therapy. Third, the current study does not distinguish between steroid doses or route of administration, while these factors could have specific effects on the results.

We believe that our findings, combined with bedside inspection of eosinophil and leukocytosis levels, may contribute to clinicians’ understanding of patient compliance to steroid therapy, as well as the understanding of the specific, distinct effect of steroids on WBC count.

## Figures and Tables

**Figure 1 jcm-08-01697-f001:**
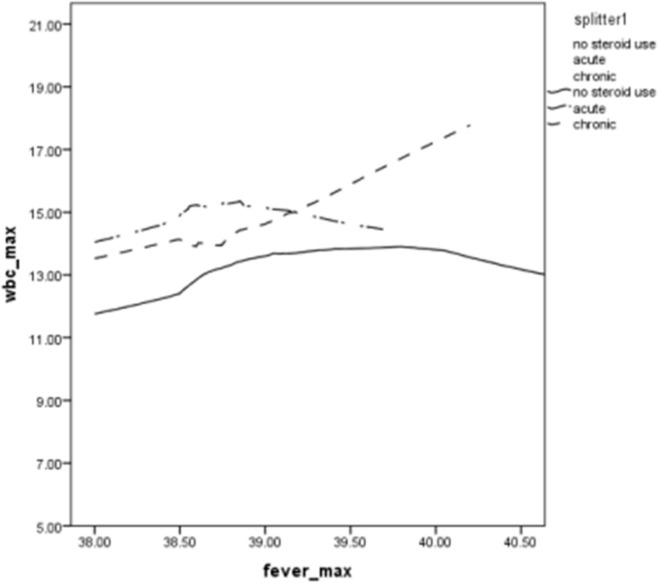
Maximum WBC count as a function of maximum body temperature for COPD patients receiving acute steroid treatment, chronic steroid treatment or no steroid treatment.

**Table 1 jcm-08-01697-t001:** Population characteristics.

	Acute Steroid Use (Under 28 Days) (*N* = 116)	Chronic Steroid Use (More than 28 Days) (*N* = 161)	No Steroid Use (*N* = 557)	*p* Value
**Age**				0.124
**Mean ± SD** **min, max**	69.40 ± 12.03 45.00, 102.00	71.74 ± 11.00 31.00, 95.00	69.48 ± 13.26 22.00, 108.00	
**Gender**				0.932
**Male % (*N*)**	61.2% (71)	63.4% (102)	62.1% (346)	
**BMI**				0.183
**Mean ± SD** **min, max**	25.20 ± 5.71 15.62, 42.97	24.82 ± 4.80 15.43, 40.25	25.88 ± 6.24 13.28, 49.95	
**Weight**				0.268
**Mean ± SD** **min, max**	70.63 ± 17.27 35.00, 115.00	70.31 ± 14.98 40.00, 120.00	72.60 ± 17.97 34.00, 150.00	
**Obesity (BMI > 30 kg/m^2^) % (*N*)**	29.6% (32)	20.4% (29)	31.3% (160)	0.040
**Smoker (current or former) % (*N*)**	74.1% (86)	77.0% (124)	69.8% (389)	0.170
**Type 2 diabetes % (*N*)**	40.5% (47)	57.8% (93)	39.0% (217)	<0.001
**HTN % (*N*)**	45.7% (53)	44.1% (71)	44.3% (247)	0.960
**Ischemic heart disease % (*N*)**	19.8% (23)	16.8% (27)	24.4% (136)	0.096
**Charlson’s comorbidity index,** **Median, (IQR)**	2, (1,3)	2, (1,2)	2 (1, 2)	0.301
**Bacteremia, % (*N*)**	2.6% (3)	1.9% (3)	4.7% (26)	0.199
**Pneumonia, % (*N*)**	37.1% (43)	33.5% (54)	37.5% (209)	0.650
**Sepsis, % (*N*)**	3.4% (4)	4.3% (7)	4.1% (23)	0.927
**Maximal fever during hospitalization,** **Mean ± SD** **min, max**	38.5 ± 0.4 38.00, 39.69	38.5 ± 0.5 38.00, 40.20	38.6 ± 0.6 38.00, 42.00	0.302
**Transfer to ICU % (*N*)**	0.9% (1)	1.2% (2)	2.3% (13)	0.452
**SLE % (*N*)**	0.9% (1)	0.0% (0)	0.0% (0)	0.045
**RA % (*N*)**	3.4% (4)	1.9% (3)	1.1% (6)	0.162

BMI: body mass index; HTN: hypertension; ICU: intensive care unit; SLE: systemic lupus erythematosus; RA: rheumatoid arthritis.

**Table 2 jcm-08-01697-t002:** Comparison of laboratory tests between the groups.

		Acute Steroid Use (Under 28 Days) (*N* = 116)	Chronic Steroid Use (More than 28 Days) (*N* = 161)	No Steroid Use (*N* = 557)	*p* Value
**Lab Tests During the First 24 h of Hospitalization** **Mean ± SD** **Min, Max**	Initial WBC *	15.36 ± 9.32 2.58, 75.48	13.90 ± 5.67 1.28, 37.40	12.34 ± 5.86 2.00, 42.10	<0.001
	WBC (mean) **	14.87 ± 8.87 2.58, 72.97	13.53 ± 5.47 1.57, 37.40	12.15 ± 5.73 2.43, 44.70	<0.001
	WBC (max) **	15.70 ± 9.51 2.58, 75.48	14.14 ± 5.70 1.85, 37.40	12.74 ± 6.07 2.43, 47.30	<0.001
	Initial Neutrophil	12.60 ± 6.49 2.13, 36.90	11.90 ± 5.40 1.20, 39.50	10.39 ± 5.61 0.97, 39.90	<0.001
	Neutrophils (mean)	12.32 ± 6.15 2.13, 31.01	11.70 ± 5.29 1.32, 35.50	10.30 ± 5.50 0.97, 42.35	<0.001
	Neutrophils (max)	13.12 ± 6.79 2,13, 36.90	12.18 ± 5.44 1.44, 35.50	10.82 ± 5.83 0.97, 44.80	<0.001
	Initial Eosinophils *	0.07 ± 0.11 0.00, 0.75	0.07 ± 0.14 0.00, 1.00	0.06 ± 0.10 0.00, 1.19	0.469
	Eosinophils (mean) **	0.06 ± 0.13 0.00, 1.08	0.07 ± 0.13 0.00, 1.00	0.06 ± 0.09 0.00, 1.19	0.517
	Eosinophils (min) **	0.05 ± 0.10 0.00, 0.75	0.05 ± 0.13 0.00, 1.00	0.04 ± 0.08 0.00, 1.19	0.508
**Lab Tests During Hospitalization, from 24 h Following Admission until Discharge** **Mean ± SD** **Min, Max**	WBC (mean)	13.93 ± 6.88 6.06, 47.53	12.51 ± 5.01 3.40, 29.69	10.77 ± 4.46 3.38, 37.97	<0.001
	WBC (max)	16.14 ± 9.84 6.32, 67.09	14.17 ± 6.92 3.40, 44.32	12.12 ± 6.22 3.39, 51.88	<0.001
	Neutrophils (mean)	11.28 ± 5.28 5.07, 31.51	10.56 ± 5.02 2.70, 28.69	8.50 ± 4.38 1.02, 35.79	0.001
	Neutrophils (max)	13.49 ± 7.86 5.07, 41.22	12.17 ± 6.85 2.70, 43.28	9.84 ± 6.13 1.02, 39.97	0.001
	Eosinophils (mean)	0.08 ± 0.09 0.00, 0.40	0.05 ± 0.07 0.00, 0.50	0.11 ± 0.14 0.00, 1.13	<0.001
	Eosinophils (min)	0.04 ± 0.05 0.00, 0.20	0.05 ± 0.07 0.00, 0.50	0.88 ± 0.14 0.00, 0.96	<0.001
**Baseline Lab Tests: Data from within 30 Days Prior to Admissions** **Mean ± SD** **Min, Max**	WBC (mean)	10.00 ± 5.47 5.62, 25.46	10.15 ± 2.85 5.59, 17.13	8.50 ± 2.55 3.77, 15,64	0.027
	WBC (max)	10.05 ± 5.62 5.62, 26.10	10.17 ± 2.83 5.59, 17.13	8.51 ± 2.57 3.77, 15.64	0.027
	Neutrophils (mean)	6.94 ± 4.79 2.70, 18.62	7.47 ± 2.84 3.38, 14.45	5.54 ± 2.47 1.93, 12.44	0.007
	Neutrophils (max)	7.07 ± 5.14 2.70, 20.30	7.50, 2.82 3.38, 14.45	5.57 ± 2.47 1.93, 12.44	0.008

* Refers to the first blood test at admissions. ** Refers to all tests during the first 24 h of admission.

**Table 3 jcm-08-01697-t003:** Linear regression model for the prediction of maximal white blood cell (WBC) count within 24 h from admissions.

Variable	Unstandardized B	Standardized B	95% CI		*p* Value
			Min	Max	
**No steroids (reference group)**	1				
**Chronic Steroids**	1.776	0.099	0.526	3.026	0.005
**Acute Steroids**	2.945	0.144	1.523	4.368	<0.001
**Age**	0.016	0.028	−0.024	0.056	0.433
**Smoker**	0.820	0.052	−0.300	1.939	0.151
**Maximal Fever**	1.113	0.085	0.207	2.020	0.016
**Pneumonia**	2.342	0.159	1.311	3.372	<0.001
**Sepsis**	4.061	0.104	1.384	6.737	0.003

**Table 4 jcm-08-01697-t004:** Linear regression model for prediction of expected WBC.

	Eosinophils = 0		Eosinophils > 0	
	Acute Steroids (46)	Chronic Steroids (72)	Acute Steroids (63)	Chronic Steroids (82)
**Difference between observed and expected** **(WBC)**				
**Mean ± SD,** **Min, Max** **Median,** **IQR**	5.06 ± 12.70 −10.71, 61.19 1.76 −3.25, 9.44	2.82 ± 7.43 −7.75, 29.02 0.51 −1.60, 5.60	1.49 ± 5.66 −10.29, 15.64 0.86 −2.93, 3.57	0.83 ± 5.65 −8.28, 19.02 0.05 −3.36, 4.20
**Difference between observed and expected** **(NEUT)**				
**Mean ± SD** **Min, Max** **Median** **IQR**	4.65 ± 9.47 −9.60, 31.11 2.37 −2.37, 9.61	3.03 ± 7.20 −7.29, 29.84 0.99 −1.30, 5.97	0.97 ± 4.99 −0.853, 12.92 0.33 −2.55, 2.41	0.86 ± 5.54 −9.88, 17.15 0.47 −3.37, 3.33

**Table 5 jcm-08-01697-t005:** Adjusted maximal WBC count.

Variable	B	β	95% CI		*p* Value
			Min	Max	
**Age**	0.034	0.069	−0.009	0.077	0.118
**Gender (Female)**	−0.508	−0.038	−1.682	0.666	0.396
**Maximal Fever**	0.994	0.086	−0.005	1.992	0.051

Linear regression model 1 in the “no-steroid group”. Adjusted R square = 0.008.

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
