# Peer review of "The Association of Glucocorticosteroid Treatment with WBC Count in Patients with COPD Exacerbation"

_jcm, 2019, doi:10.3390/jcm8101697_

Reviewer 1 Report

This interesting study tries to quantify and compare the effect of acute/ chronic steroid administration versus no steroid administration on the WBC and neutrophil count in patients with COPD exacerbation and fever. Eosinophil count was used as a marker for steroid exposure. It was shown that patients with lower eosinophil counts had a larger effect on total WBC and neutrophil counts and this should be taken into consideration when evaluating patients with COPD exacerbations and fever.

Comments:

Introduction should include an explanation regarding the logic for the use of the eosinophil count as a marker for steroid use in clinical practice Methods - Please define the study population - how is COPD diagnosed? by whom?  Methods - Steroid use: Is there any information on oral versus IV ? Mean doses?   Ethics - Please provide ethics approval number Results: Table 1:Please explain the low minimal ages (31,22). Were these patients actually diagnosed with COPD? Were patients with asthma excluded? How was obesity defined? Is the co-morbidity index performed for all patients admitted to the hospital? SLE  and RA patient- This is a small subgroup of patients - why are they mentioned? are they not immunosuppressed by other mechanisms? Table 2 - Please clarify table. What is the difference between initial WBC mean?  from admission WBC means?  Baseline lab tests - were these available for all patients? how far back from admission? were they associated with prior admissions or medical events? Table 3 - Please explain the regression model, include only significant parameters and provide R square Table 4 - Are the expected WBC and NEUT obtained for admission values? mean values during admission?  Please remove either figure 1 or 2. Discussion line 153 - There needs to be some reservation to this recommendation and clarification that it does not necessarily imply that the patient should not receive antibiotic therapy! Discussion line 167- Please add other possible causes for a normal or high level of eosinophils  Limitations - please address the issue of including all steroid doses and methods of administration in one dichotomal value

Reviewer 2 Report

The study investigated the association of glucocorticosteroid treatment with WBC Count in Patients with COPD Exacerbation, which is an overall good study. Here are a few questions the authors need to clarify.

Major:

Please emphasize the novelty of this study in both introduction and discussion. lots of relevant studies (leukocytosis in COPD) are not cited in this study. Eg. Frenkel A, Estimations of a degree of steroid-induced leukocytosis in patients with acute infections. Am J Emerg Med. 2018. The description of Figure 1&2 are not clear and please clarify how they were drawn.

Minor:

Please unify the font used in the text. Eg. Text in the 3.2.
